

# Cross-modal and subliminal effects of smell and color

Naoto Sato[1,2], Mana Miyamoto[1,3], Risa Santa[1], Ayaka Sasaki[3] and Kenichi Shibuya[1,3]

[1] Graduate School of Health and Welfare, Niigata University of Health and Welfare, Niigata, Japan
[2] Department of Health and Nutrition, Yamagata Prefectural Yonezawa University of Nutrition Sciences, Yonezawa, Japan
[3] Department of Health and Nutrition, Niigata University of Health and Welfare, Niigata, Japan

## ABSTRACT

In the present study, we examined whether the cross-modal effect can be obtained between odors and colors, which has been confirmed under olfactory recognizable conditions and also occurs under unrecognizable conditions. We used two flavors of red fruits such as strawberries and tomatoes for this purpose. We also aimed to compare whether similar cross-modal effects could be achieved by setting the flavors at recognizable (liminal) and unrecognizable (subliminal) concentrations in the experiment. One flavor at a normal concentration (0.1%, Liminal condition) and one at a concentration below the subliminal threshold (0.015%, Subliminal condition), were presented, and the color that resembled the smell most closely from among the 10 colors, was selected by participants. Except for the subliminal tomato condition, each odor was significantly associated with at least one color ($p < 0.01$). Participants selected pink and red for liminal strawberry (0.1%) ($p < 0.05$), pink for subliminal strawberry (0.015%) ($p < 0.05$), and orange for liminal tomato (0.1%) ($p < 0.05$), but there was no color selected for subliminal tomato (0.015%) ($p < 0.05$). The results of this study suggest that the flavor of tomato produced a cross-modal effect in liminal conditions, but not in subliminal conditions. On the other hand, the results of the present study suggest that the flavor of strawberries produces a cross-modal effect even under subliminal conditions. This study showed that cross-modal effects might exist, even at unrecognizable levels of flavor.

## INTRODUCTION

The understanding of how the brain processes, integrates, and perceives external stimuli is of great importance. Cross-modal phenomena have been widely studied, in which perceptions that are typically separate, such as taste and vision, or vision and smell, influence each other. For example, individuals tend to associate certain colors with specific smells, with the smell of lemons often being described as "yellow" and the smell of ripe strawberries being associated with the color "red". This phenomenon, known as a cross-modal effect, has been demonstrated through numerous studies (*Heatherly et al., 2019*; *Kemp & Gilbert, 1997*; *Luisa Demattè, Sanabria & Spence, 2006*; *Sheifferstein & Howell, 2015*; *Sheifferstein &*

Corresponding author
Kenichi Shibuya, shibuya@nuhw.ac.jp

*Tanudjaja, 2004*). *Gottfried & Dolan (2003)* showed that the presentation of visual stimuli with complex pictures may affect olfactory information processing. In addition, *Morrot, Brochet & Dubourdieu (2001)* showed an association between odor and color, as white wine artificially colored red with odorless dyes was determined to be red wine based on visual and olfactory information. The study by *Luisa Demattè, Sanabria & Spence (2006)* has shown a significant relationship between odor and color, with caramel odors being associated with the colors "brown" or "yellow", cucumber odors being associated with the color "green", and strawberry odors being associated with the colors "pink" or "red". These findings are consistent with the general colors of foodstuffs and further support the existence of strong cross-modal associations between vision and smell.

Subliminal effects refer to the phenomenon of unconsciously stimulating an individual using sub-threshold, or subliminal, stimuli (*Yamada et al., 2014*). One example of this is the repeated use of subliminal speed and volume advertisements on television and radio, which can increase viewers' desire to purchase a product. These effects can influence actions, choices, and thoughts, and have been demonstrated in several studies. For example, *Holland, Hendriks & Aarts (2005)* found a significant difference in word selection for cleaning tasks when participants were presented with a citrus scent under the subliminal threshold, leading to changes in social liking of human facial expressions. These studies demonstrate that olfactory information can modulate human cognition and behavior even when presented at subliminal levels. In addition, the study by *Li et al. (2007)* provided evidence for the influence of olfactory information on human cognition and behavior, even when presented at subliminal levels. *Li et al. (2007)* utilized a facial expression judgment task and presented participants with pleasant, unpleasant, and neutral odors under subliminal thresholds. The results of this research indicated that this manipulation significantly affected the social liking of human facial expressions. Previous studies have shown that olfactory information can modulate human cognition and behavior even when odor stimuli are presented under a subliminal threshold.

While both cross-modal and subliminal effects have been studied in various ways, few studies have examined cross-modal effects under subliminal conditions, and there are many unknowns and a lack of knowledge in the field of multisensory integration (*Stein & Stanford, 2008*). In particular, the cross-modal link between vision and olfaction has received limited attention. *Yamada et al. (2014)* have shown that odors presented under subliminal conditions can increase preference for unfamiliar food images. This effect of odor in food preference may be related to food neophobia (*Pliner & Salvy, 2006*). Food neophobia is defined as a feeding avoidance response to novel foods, and the study by *Raudenbush et al. (1998)* has found that individuals with food neophobia react and evaluate odors differently than those without. Furthermore, odor control may reduce stress over unfamiliar foods. Food neophobia can have a negative impact on health in terms of nutrition, as rejecting novel foods can narrow the range of foods consumed. For example, food neophobia has been found to be negatively correlated with vegetable consumption in 7-year-old girls (*Galloway, Lee & Birch, 2003*) and with fruit and vegetable consumption in preschool children (*Cooke et al., 2004*). Considering this, the present study aims to examine the cross-modal effects of vision on food odors.

The present study aims to examine the cross-modal effect between olfaction and vision using two popular red agricultural products in Japan, tomatoes and strawberries, as olfactory stimuli. Tomatoes and strawberries were chosen as stimuli due to their distinct aromatic components. The aromatic components of tomatoes include cis-3-hexenal and trans-2-hexenal (*Wang & Seymour, 2017*), which are commonly found in green fruits such as cucumbers and cabbage, while the aromatic components of strawberries are primarily composed of alcohols, esters, and aldehydes, with furaneol being the main aromatic component (*Schwab, 2013*). The specific aromatic component of strawberries may make it more likely for the odor and color of strawberries to be easily recognized in association with one another. The present study will examine whether different reactions are observed depending on the two red fruits, one with an odor that explicitly evokes an image (strawberries) and the other with an odor that evokes several fruits (tomatoes) when stimuli are set at recognizable (liminal) and unrecognizable (subliminal) concentrations. The present study also aims to examine the relationship between odor and color. The hypotheses of the present study are that olfactory information will connect to the visual information regardless of the subject's awareness of the odors and that the olfactory information is utilized only for the subliminal odors that evoke an image of the object.

## METHODS

### Participants

The study participants consisted of 39 healthy people, between 18 and 22 years of age and with normal or corrected-to-normal eyesight. They received an explanation regarding their written informed consent, which included a description of the risks of the experiment. The Ethics Committee of the Niigata University of Health and Welfare approved this study (Approval number: 18243-190705). All the participants were naïve regarding the study protocol. The research contents were explained verbally to all participants. Written informed consent was obtained from each participant after a full explanation of the nature of the study procedure and its non-invasiveness. The number of participants was calculated by G*Power (*Faul et al., 2007*) prior to the experiment (Effect size = 1.2, alpha = 0.005, $df = 19$) determined from the results of preliminary experiments.

### Protocol

Upon their arrival at the laboratory, participants were asked to sit in front of a computer in a small room for 15 min, to create a resting state. They were assured that their responses would remain completely anonymous as ID numbers were used to manage the data. They were then given instructions regarding the tasks conducted in the experiment. The experimenter then left the room to ensure the participants' anonymity during the experiment. The participants performed the tasks while facing the computer. All tasks were performed using a computer program written in PsychoPy (*Peirce, 2007*; *Peirce, 2009*; *Peirce et al., 2019*). The stimuli were presented on a 17″ (inch) CRT monitor (LCS 172VXL; NEC, Japan) with a resolution of 1,024 × 768 pixels and a refresh rate of 100 Hz. Stimulus presentation and data collection were controlled using a computer (M8-D; NEC, Tokyo, Japan). Stimuli were presented at a viewing distance of 40 cm. The luminance of the fixation

point was 91.0 cd/m$^2$. The command cursors were the white boxes surrounding each rating value (0.95 × 1.89°; 91.0 cd/m$^2$); selected boxes were filled in with white color.

The participants were tested under two conditions: (1) flavors with liminal concentration and (2) flavors with subliminal concentration. Two flavors were used: strawberries (T & M Co., Ltd.) and tomatoes (Yokoyama Flavoring Co., Ltd.). Each flavor was diluted with ion-exchanged water to achieve liminal (0.1%) and subliminal concentrations (0.015%) (*Yamada et al., 2014*) to create four different samples. The four flavors were correctly identified by the participants as liminal and subliminal after all experiments. Odorless air was released with an aroma diffuser (Muji Supersonic Wave Aroma Diffuser; Ryohin Keikaku Co., Ltd.) from 1–2 h before the experiment and continued until the end of the experiment. In addition, a drop of the sample was placed on an aroma-testing paper and the paper was placed three cm away from the subject's nose for 4 s. The participants were wearing blindfolds and ear-muffs, during the experiment as in a previous study (*Yamada et al., 2014*).

All ten colors (red, yellow, blue, green, orange, pink, brown, turquoise, purple, and gray) were displayed on the monitor, and the color that felt closest to their mood was selected by participants, as in a previous study (*Luisa Demattè, Sanabria & Spence, 2006*). Participants came to the laboratory eight times to be tested. The time between each visit was set to be open for at least 24 h. The same procedure was repeated twice for four different samples. The order of the ten colors displayed on the monitor for 15 s and the order of smelling the samples were randomized. The participants were not informed of the flavor of the odor used for the samples. The RGB values for the 10 colors were as follows: red (231, 0, 0), yellow (248, 248, 0), blue (0, 48, 255), green (0, 85, 0), orange (255, 85, 12), pink (255, 0, 193), brown (98, 48, 0), turquoise (0, 229, 189), purple (99, 13, 253), and ash (56, 56, 56) (Fig. 1). Participants' judgments were saved in csv files. The colors, and odor types selected by the participants were statistically evaluated by residue analysis using the chi-square test (*McHugh, 2013*) for the colors associated with each odor type.

After the experimental blocks, participants were asked whether they were aware of the odor and of the influence of the odor on their performance (*Holland, Hendriks & Aarts, 2005*). The interview after all the experimental blocks showed that none of the participants were aware of the odor in the subliminal conditions and all the participants were aware of and identified the odor in the liminal conditions.

### Statistical analysis

We used a nonparametric $\chi^2$ statistic to test whether the observed color associations for each of the odors deviated from an equal distribution. Given that we were interested in possible associations between particular odors and specific colors, we conducted post-hoc $\chi2$ comparison using residual analysis (Bonferroni corrected, $\alpha = 0.005$ (0.05/10 colors)) between the participant's responses to the colors for each odor.

## RESULTS

The mean age of the participants was 19.7 ±0.8 years. The chi-square test was used to compare the frequency of each odor with the color selected by the subjects, statistical

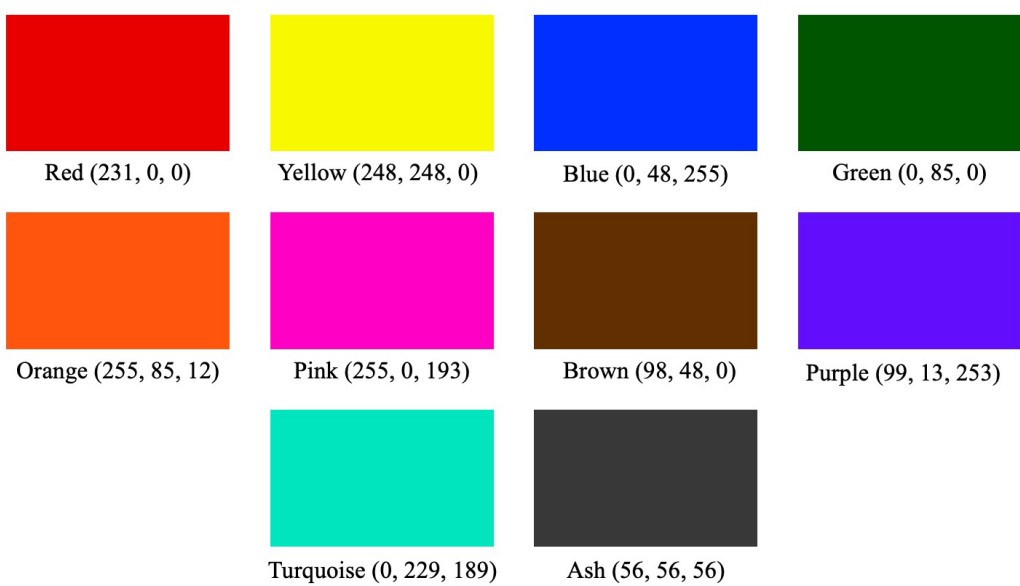

**Figure 1** **The task used in the experiment.** Participants should judge which of the given colors on the screen can best match the odor they smell, by clicking the corresponding color square directly when they decide to choose. Participants responded with the mouse.

analyses were performed, and the results were evaluated by residue analysis (Table 1). The results of the $\chi 2$ analysis showed that subjects' color responses to a given odor were not equally distributed among the 10 color options (*i.e.,* they responded to a particular color on significantly more than 10% of trials) (upper row of Table 1, Liminal Strawberry; $p < 2.72 \times 10-11$, Subliminal Strawberry; $p < 1.30 \times 10-5$, Liminal Tomato; $p < 8.80 \times 10-3$). Most of the odors used in the experiment were significantly ($p < 0.005$) associated with at least one color (see a lower row of Table 1 for the respective $p$ values). As in Tables 1 and 2, the colors significantly selected for the liminal (0.1% concentration) strawberry flavor were pink (50.0%) and red (23.7%). The color of the subliminal (0.015% concentration) strawberry was pink (42.1%). The most significantly selected color for the liminal (0.1% concentration) tomato flavor was orange (31.6%). No significant difference was observed for any color in the subliminal (0.015% concentration) tomato flavor.

## DISCUSSION

The data analysis of the present study reveals that the flavors of foods, whether presented at liminal or subliminal concentrations, produce a cross-modal effect and evoke common colors. Specifically, our results show that a characteristic flavor, such as the flavor of strawberries, whether presented at liminal or subliminal concentrations, evokes a similar color to that of the food. The color "pink" was significantly selected for both liminal (0.1% concentration) and subliminal (0.015% concentration) strawberry flavors. This finding supports the previous studies of *Yamada et al. (2014)*, who found that visual choices were influenced by both conscious and unconscious olfactory processing. Additionally, research has repeatedly shown that cognitive processing can occur unconsciously (*Tsuchiya &*

**Table 1** The *p*-values and effect sizes for residue analysis.

| | Strawberry | | Tomato | |
|---|---|---|---|---|
| | Liminal | Subliminal | Liminal | Subliminal |
| $\chi^2$ | 90.5 | 56.6 | 36.6 | 11.0 |
| df | 19 | 19 | 19 | 19 |
| *p* | $2.72 \times 10^{-11}$ | $1.30 \times 10^{-5}$ | $8.80 \times 10^{-3}$ | 0.924 |
| Effect size w | 1.52 | 1.21 | 0.97 | 0.53 |
| Red | 0.001[*] | 0.311 | 0.262 | 0.311 |
| Yellow | 0.037 | 0.262 | 0.262 | 0.957 |
| Blue | 0.037 | 0.122 | 0.037 | 0.311 |
| Green | 0.037 | 0.037 | 0.311 | 0.957 |
| Orange | 0.311 | 0.557 | 0.000[*] | 0.557 |
| Pink | 0.000[*] | 0.000[*] | 0.631 | 0.957 |
| Brown | 0.311 | 0.311 | 0.311 | 0.311 |
| Turquoise | 0.037 | 0.122 | 0.122 | 0.311 |
| Purple | 0.262 | 0.631 | 0.262 | 0.557 |
| Gray | 0.037 | 0.311 | 0.037 | 0.007 |

Notes.

*Results of χ2 analysis for each of the odors are reported in the upper and lower rows. Significant *p*-values in the upper and lower rows mean that participants' color responses to a given odor were not equally distributed among the 10 color options (*i.e.*, they responded to a particular color on significantly more than 10% of the trials). The lower rows show the results of the post hoc comparisons (Bonferroni corrected, $\alpha = 0.005$ (0.05/10 colors) used to detect responses to a particular color, along with the percentage of each color choice.

**Table 2** Odors and related colors.

| Odors | Related colors |
|---|---|
| Strawberry (0.1%) | Pink (50.0%), Red (23.7%) |
| Strawberry (0.015%) | Pink (42.1%) |
| Tomato (0.1%) | Orange (31.6%) |
| Tomato (0.015%) | None |

*Adolphs, 2007*) and that odors can influence visual processing, even when presented at subliminal levels (*Holland, Hendriks & Aarts, 2005*). The study by *Luisa Demattè, Sanabria & Spence (2006)* also reported a cross-modal association between odor and color in liminal strawberry flavor. The present study extended previous research by demonstrating that this cross-modal effect between odor and color was obtained in strawberries under both liminal and subliminal conditions. However, in the case of tomatoes, which share common odors with other agricultural products such as cucumbers, a cross-modal effect between color and odor was only obtained under liminal conditions and not under subliminal conditions. This result is consistent with the findings of previous neurophysiological studies (*Barutchu, Spence & Humphereys, 2018*), which have shown that cross-modal effects can occur even in subliminal stimuli.

The results of the present study suggest that strawberries are generally perceived as red and pink, as there was a significant difference in the selection of these colors between varieties in the experiment. Participants tended to select pink more frequently than red,

which is consistent with previous studies by *Luisa Demattè, Sanabria & Spence (2006)*. One possible explanation for this is that the strong association between strawberries and pink, which is commonly used in packaging design and food products associated with the flavor and aroma of strawberries in everyday life, may have influenced the results of this experiment. Additionally, the significant selection of pink over red may also be since many foods marketed as strawberry-flavored use more pink than red in their packaging design or in the color of the food itself.

The results of the present study suggest that even at the subliminal concentrations (0.015% concentration), a cross-modal effect was observed between the odor of strawberries and the color pink. However, the color red, which was significantly chosen for the liminal strawberry flavor, was not significantly chosen for the subliminal strawberry flavor, indicating that the cross-modal effect was not as strong for the subliminal strawberry flavor as it was for the liminal strawberry flavors. In contrast, no colors were significantly selected for subliminal tomato flavors, which suggests that tomato flavors do not achieve the same cross-modal effects at subliminal and liminal concentrations. This is consistent with previous research by *Yamada et al. (2014)*, which found that subliminal tomatoes were more difficult to judge than subliminal strawberries. It is possible that the subliminal tomato flavor was more difficult to judge in the current study, which could have led to variations in the participants' color choices.

It is well known that multisensory integration is stronger when the information reaching multiple sensory receptors originates from the same, rather than a different, spatial region in the brain (*Stein & Meredith, 1993*; *Meredith et al., 2022*). In our experiment, the odors reached the olfactory cortex of the participants and were transmitted to the hippocampus, which recalled the odors of the past. The hippocampus also contains memories of food that are strongly bound to the odor, making it likely that participants can recall the food and its colors from the odor. In contrast, in our experiment, orange was significantly selected for the liminal (0.1% concentration) tomato flavor. While most tomatoes are red, there are also yellow and orange varieties, which may explain why the significant difference was orange rather than red. It is possible that participants' perception of the odor of tomatoes depends on their accumulated experience. Additionally, participants typically reported that in the liminal strawberries, but could not tell at all in other trials (liminal tomato, subliminal tomato, and subliminal strawberries) due to the random repetition of stimuli. This suggests that the choice of color for the liminal tomato odor may have been made unconsciously, resulting in a less vivid choice of color. At this moment, there are no previous studies that can justify the choice of orange to tomato flavor. However, it is unlikely that the participants have a particularly biased opinion about the color of tomatoes. Future studies will justify the reasons for these results.

The findings of the present study suggest that the cross-modal effect under subliminal conditions is flavor-specific. The aromatic components of tomatoes include cis-3-hexenal and trans-2-hexenal (*Wang & Seymour, 2017*), which are not specific to tomatoes by are also found in other fruits such as cucumbers, and cabbage. This may explain the variation in color selection observed in the present study. In contrast, the aromatic components of strawberries include alcohols, esters, and aldehydes, with furaneol as the main aromatic

component (*Schwab, 2013*). Since strawberries have a specific aromatic component, it is possible that the odor and color of strawberries are easily recognized in association with strawberries. However, tomatoes lack a specific aromatic component, which may have made it difficult for tomatoes to be recognized in association with their odor and color. These findings suggest that the cross-modal effect under subliminal conditions is dependent on the specific flavor being presented.

## CONCLUSION

In conclusion, the present study found that a cross-modal effect of odor and color was achieved for both liminal (0.1% concentration) strawberry and tomato flavors. The results showed that the liminal strawberry flavor evoked a pink color while the liminal strawberry flavor evoked an orange color. Furthermore, the study also found that the cross-modal effect between odor and color was achieved under subliminal conditions specifically odor, but not for tomato odor. These findings highlighted the importance of flavor-specificity in cross-modal effects, and the need for further research to understand the underlying mechanisms of multisensory integration under subliminal conditions.

### Funding

The authors received no funding for this work.

### Competing Interests

Kenichi Shibuya is an Academic Editor for PeerJ.

### Author Contributions

- Naoto Sato conceived and designed the experiments, analyzed the data, authored or reviewed drafts of the article, and approved the final draft.
- Mana Miyamoto conceived and designed the experiments, authored or reviewed drafts of the article, and approved the final draft.
- Risa Santa conceived and designed the experiments, authored or reviewed drafts of the article, and approved the final draft.
- Ayaka Sasaki conceived and designed the experiments, performed the experiments, prepared figures and/or tables, authored or reviewed drafts of the article, and approved the final draft.
- Kenichi Shibuya conceived and designed the experiments, analyzed the data, prepared figures and/or tables, authored or reviewed drafts of the article, and approved the final draft.

### Human Ethics

The following information was supplied relating to ethical approvals (i.e., approving body and any reference numbers):

The Ethics Committee of the Niigata University of Health and Welfare approved this study (Approval number: 18243-190705).

## Data Availability

The raw measurements are available in the Supplementary File.

## Supplemental Information

Supplemental information for this article can be found online at http://dx.doi.org/10.7717/peerj.14874#supplemental-information.

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
