# Peer review of "Cross-modal and subliminal effects of smell and color"

_PeerJ, doi:10.7717/peerj.14874_

## Round 0.1 · original submission · Major Revisions

The two authors raise major questions and concerns, which I believe will need to be fully addressed before acceptance.

Reviewer 1 ·

Basic reporting

Inspired by the literature on cross-modal integration, Sato and colleagues examined whether the exposure to either liminal or subliminal odors led participants to select a given color more frequently to the detriment of other colors not related to the odor to which they were exposed. The authors reported that strawberry flavors ‘produce a cross modal-effect even under subliminal conditions, while tomato flavor produced a cross-modal effect only in liminal conditions.
Although I find the idea interesting, the research article lacks information I consider relevant to understand the authors’ main findings as well as their validity. Moreover, in my perspective, some methodological problems may result in confounds precluding the authors to make a strong case from their data. Below, and across the several sections included in this review platform, I enumerate some of them.

For instance, in the introductory section, although the reported study is on the main field of cross-modal integration and multi-sensory integration, the authors only briefly mentioned this research field. In my opinion, a more extensive literature review would benefit the paper, making clear some of the claims the authors made in the introductory section (e.g., see Stein & Stanford, 2008, Nature Reviews Neuroscience).

Moreover, clear research hypotheses are missing, impairing the understanding of the nature of the authors’ research, precluding a clear understanding of the obtained results, and making it hard to identify the gap the present research is fulfilling.

Experimental design

Additionally, in the methodological section, from my perspective, more details and additional control are needed. For instance, it is not clear whether the procedure used to confirm that the used flavors were in fact subliminal or liminal. No data is reported regarding that matter. Moreover, were the flavor concentrations calibrated per participant? How can the authors ensure that their participants have identical smell capacities (no smell capacity assessment is reported)? These methodological procedures or a clear justification for their absence should be reported in most of the research involving chemical senses, especially after a pandemic situation that resulted in several olfactory and taste impairments.
It would also be relevant to have more details regarding the sample size estimation. Taking into account the design of the study (which should also be clearly stated), I suspect that more participants are needed to achieve power (e.g., Brysbaert, 2019, Journal of Cognition).

Moreover, additional details are needed regarding the data analyses. What is a residue analysis? Is this suitable for this type of data? Reporting test statistics (e.g., the Chi-squared test value) and the degrees of freedom would facilitate the results' interpretation. Additionally, what pairwise comparisons were performed? Were they corrected? These are all details that, in my opinion, would ease the interpretation and assessment of the reported results.

Validity of the findings

From my perspective, the obtained results should also be further discussed and integrated with the existing literature. For instance, is there any data regarding the tomato flavor that justifies its association with the orange color and not red,? How can this be related to the specific population in which the study was conducted? Limitations and direction for future studies would also be valuable.

Reviewer 2 ·

Basic reporting

First of all, I would like to appreciate the authors for their effort in this research. Although written well, there are a lot of revisions that are needed to improve the paper to publishable standards. I would encourage the authors to carefully make those changes and give clarity to the readers. Here are a few of my comments.
- Tomatoes and Strawberries are fruits according to much literature. But it is reported as vegetables in this paper. Could you mention why and with evidence?
- There is a lack of literature citing in the totality of the paper.
_ There is no novelty in the experiment and the research questions/hypothesis are not clearly mentioned.
- There is a lack of clarity in mentioning why this research is important.

Experimental design

The section was well written.
- Line 94 mentioned that the resting time exceeds 15 minutes. The resting time has to be specifically mentioned. It means that people were made to wait anywhere from 15 minutes to an infinite number of hours. Please change that.
- How were the scents selected? What scents were used?
- Why were tomatoes and strawberries chosen for the main objects of study?
Why not other fruits?
- Why are all the colors chosen? Is there a standard protocol for that?
- How was the odorless air pumped into the room?
- How were the liminal and subliminal concentrations selected?

Validity of the findings

From table 1, it is seen that for liminal strawberries there are significant differences in the association with yellow, blue, green, turquoise, and gray colors as well other than just red and pink. There was no mention of that in the results section.
- There needs to be an explanation of the results a bit more elaborately.
- A picture or illustration of the experimental setup would be helpful if added.

Additional comments

Line 172 - However, no significant... - This line is confusing and not easy to understand. Please rewrite this sentence.

Table 1 title spelling error.. Effect sizes for "residue" analysis , not redidue analysis
Table 1 should have an indication of which p-values are significant.

The authors should take more caution in framing sentences and also referencing them to previous literature. Explaining the comments would improve the paper substantially. At present, I am not able to see the importance and contribution of this paper clearly. All the very best to the authors. I look forward to hearing the rebuttal if allowed.

---

## Round 0.2 · Major Revisions

Dear authors,

Please fully address the comments made by reviewer number 2.

Reviewer 2 ·

Basic reporting

I still don't see the importance of this research and its contribution to the scientific community in the revised version as well.
I also feel that the literature review section should be improved.

Experimental design

Good job on making new changes to the methods section.

Validity of the findings

I still feel the results should be better explain than just adding footnotes to the Table 1.

Additional comments

Line 73 - Tomatoes and strawberries... That line doesn't make sense

---

## Round 0.3 · Minor Revisions

I wish to acknowledge that the authors answered most of the reviewers questions. However, I must recommend a careful proofreading of the manuscript (e.g., "While both cross-modal and subliminal effects have been extensively studied, fewer studies focused on the cross-modal effects under subliminal conditions, or even at multisensory integration as a whole (Stein & Stanford, 2008). In this sense, it is worth examining the cross-modal link between the visual and olfactory systems. For instance, Yamada et al. (2014) showed that odors increased preference for unfamiliar food images, under subliminal conditions. This effect of odor in food preference can be extremely relevant. Food neophobia (Pliner & Salvy, 2006) is a feeding avoidance response to novel foods. Participants with food neophobia have been shown to react and evaluate odors differently than those without (Randenbush, 1998)."

---

## Round 0.4 · accepted · Accept

The authors addressed all the comments.